# A 3D Surface Plot for the Effective Visualization of Specific Serum Antibody Binding Properties

**DOI:** 10.3390/antib14030068

**Published:** 2025-08-13

**Authors:** József Prechl, Ágnes Kovács, Krisztián Papp, Zoltán Hérincs, Tamás Pfeil

**Affiliations:** 1R&D Laboratory, Diagnosticum Zrt, 1047 Budapest, Hungary; 2Department of Biostatistics, University of Veterinary Medicine Budapest, 1078 Budapest, Hungary; 3Department of Applied Analysis and Computational Mathematics, Eötvös Loránd University, 1117 Budapest, Hungary; tamas.pfeil@ttk.elte.hu; 4HUN-REN-ELTE Numerical Analysis and Large Networks Research Group, 1117 Budapest, Hungary

**Keywords:** antibody, serology, quantitative systems biology, visualization

## Abstract

Background: When an antigen molecule is exposed to serum, many different kinds of antibodies bind to it. The complexity of these binding events is only poorly characterized by assays that generate a single variable, generally reflecting the fractional saturation of the antigen, as the readout. Methods: We have previously devised an assay that delivers the essential biochemical variables to determine fractional saturation as the output: an equilibrium dissociation constant for affinity, the ratio of antibody concentration to the equilibrium constant and the concentration of bound antibodies under reference conditions. Here we propose a visualization method for the practical and informative display of these variables. Results: Using total antigen concentration and free and bound antibody concentration as coordinates in a three-dimensional space, a surface plot can depict the behavior of serum antibodies in the measurement range and identify the values of the key variables of binding activity. This surface display (antibody binding in 3-concentration display, Ab3cD) was used for the characterization of antibody binding to the SARS-CoV-2 spike protein in seronegative and seropositive sera. We demonstrate that this visualization scheme is suitable for presenting both individual and group differences and that epitope density changes, not commonly measured by immunoassays, are also revealed by the method. Conclusions: We recommend the use of 3D visualization whenever detailed, informative and characteristic differences in serum antibody reactivity are studied.

## 1. Introduction

Detection and characterization of antigen-specific antibodies from serum is one of the building blocks of immunodiagnostics. Depending on the nature and role of the antigen used, specific antibody measurement contributes to the in vitro diagnostics of infectious diseases [1], autoimmunity [2], allergies [3] and tumors [4,5]. Beyond helping to set up a diagnosis by using panels of antigens and selecting the measured antibody isotype, staging and classification of the disease state and selection and monitoring of therapy are also helped by antibody serology [6,7,8,9]. The methods of specific serum antibody detection range from point-of-care qualitative assays through standardized automated assays to quantitative tests. Most of these assays, even the most sophisticated ones, deliver results in arbitrary units [10] or expressed as equivalents of monoclonal antibody reference standards [11,12].

If we wish to evaluate the reactivity of serum antibodies, we actually should first understand what the best readout for the binding activity of specific antibody is in terms of being correlated to the clinical observation. The vast majority of assays practically measure the *number of antigen-bound antibody molecules* and are therefore related to the fractional occupancy of the target antigen: the percentage of antigen molecules that is bound by antibodies. Besides the conventional indirect detection by labeled reagents, quantitative mass spectrometry can directly identify and quantify the bound antibody molecules [13]. Fractional occupancy is in turn determined by the *affinity* (strictly speaking average or aggregate affinity) and *concentration* (average concentration) of antibodies. Affinity is related to the free energy change accompanying binding and can be characterized by the ratio of concentrations of reactants of the binding reaction to products of the binding reaction under equilibrium conditions. This value, the dissociation constant or KD, is lower for higher affinity reactions, since the bound form (products) are thermodynamically preferred to the unbound forms (reactants). The binding reaction reaches equilibrium when products and reactants are generated at the same rate and their concentrations remain constant. At this point the KD can be calculated from the respective concentrations. The difficulty of estimating KD and concentrations for serum antibodies arises from the fact that we are dealing with an immensely diverse population of antibody molecules, where both KD and concentration values should somehow characterize the whole population. Conventional common assays (ELISA, SPR) can measure these latter properties directly but separately [14,15,16,17,18,19], while novel technologies, such as microscale thermophoresis [20] and microfluidic affinity profiling [21] are being applied to address them simultaneously in a single assay.

An important and often neglected aspect of the polyclonal response against antigens is that various epitopes on the same antigen molecule may be targeted by distinct antibody clonotypes. This is one of the reasons why a monoclonal antibody may not be optimal for reference in binding assays: an increased polyclonal binding signal might be caused both by additional antibodies binding to the same antigen molecule (increased *molecular epitope density*) and by antibodies binding to additional antigen molecules (higher fractional occupancy). It is important to stress that changes in epitope density are brought about by the emergence of antibody clones and their differentiation into antibody-producing cells; therefore it is also a property of the antibody response. As long as we separately measure either the number of bound antibody molecules or their affinity and concentration, we will not be able to recognize and determine changes in molecular epitope density from changes in fractional occupancy (Figure 1).

Serum antibody measurements can be adjusted to focus on a binding property that is most relevant to us by choosing a proper method and readout. The best readout of an antibody assay is dependent on the clinical question being asked: for example, in the case of infections, it is the correlation with protectivity; in autoimmunity, it is the correlation with disease severity and for tumor diagnostics, the best sensitivity could pinpoint a good methodological approach. But already to reveal such correlations we need to be able to measure and understand the above key properties of the antibody response. Previously we showed that antibody serology results can be obtained in universal biochemical units [22,23]; now we provide a visualization method that highlights immunologically meaningful changes in reactivity and helps in the interpretation of quantitative antibody measurements. Here our goal is to identify and provide the proof-of-concept for a visualization approach that is intuitive, as simple as possible and yet reveals all the determinants of the strength of the examined antigen-specific response, to devise a plot that reflects changes in key variable values as modeled by the Richards function [24].

## 2. Methods

### 2.1. Dataset of SARS-CoV-2 Specific Antibody Binding Results

We used previously published data [23] obtained from commercially available serum samples from confirmed COVID-19-positive and -negative subjects (RayBiotech CoV-PosSet-2). Tables containing the visualized dataset are available as Appendix A. The variables used for visualization were *logC*, ν and *logx_i_* and were obtained as previously described [22,23].

### 2.2. Visualization Program

For displaying a simple 3D surface based on functions and variable values contained in a spreadsheet, we used the ‘persp’ command of the open-source free R programming language (R4.4.3).

The surface was defined by Equation (1), which defines the surface coordinates in an x-y-z Cartesian space. The x axis is the logarithm of total molar Ag concentration in the microspot, the y axis is the logarithm of serum Ab concentration, and the z axis is the logarithm of bound Ab concentration. The values of x and y are determined by the experimental settings; z is calculated from calibration curves used in the assay. In our model, bound Ab concentration is a generalized logistic function or Richards function of logarithms of Ag and Ab concentrations. The use of Richards function, compared to the logistic function, allows for non-ideality in the binding curve that appears as a shift in the inflection point from the mid-point value. This shift is captured by the asymmetry parameter ν of the Richards function. To normalize variation over a wide measurement range we use the logarithmic forms of the Richards function and display the results in logarithmic scales [22,23]. In the equation, *logC* is the logarithm of the fluorescent signal at the point where both coordinates are the inflexion points of the respective Richards functions; ν is the asymmetry parameter representing a proportionality factor of apparent and true dissociation constants; *x* is the molar concentration of immobilized antigens; x_i_ is antigen concentration at the point of inflection of the corresponding Richards curve and *y* is the relative serum antibody concentration expressed as 1/dilution factor.(1)$$z=logC+\frac{\nu }{1-\nu }*log\left(\frac{1/\nu }{1+e^{-logy}}\right)+\frac{1}{\nu }*log\left(\frac{1+\nu }{1+\nu *e^{-\left(logx-logxi\right)}}\right)$$

The code lines for the visualization are available as Appendix A.

## 3. Results

### 3.1. Visual Interpretation of Key Variables of Serum Antibody Binding

As discussed above, the three key variables that determine the number of bound antibody molecules per unit area (surface concentration) are the equilibrium dissociation constant (KD), relative antibody concentration ([Ab]/KD) and epitope density. We can probe the first two by changing the number of immobilized antigen molecules per unit area and by diluting the serum, respectively. Titration of these attributes identifies special values along the two axes, which correspond to the inflection points of titration curves when displayed on a linear scale. The reason we use a logarithmic scale is to be able to visualize events over a very wide range of values. Once we obtain these special values, namely the apparent KD derived from surface antigen concentration and the apparent KD derived from relative serum concentration, we can generate a 3D surface using the function used for fitting, Equation (1) (Figure 2). The location along the z axis of this surface is determined by the molecular epitope density, in addition to the previous two variables, KD and [Ab]. Thus, the z axis is practically the logarithm of the concentration of antigen-bound antibodies, which accommodates changes in KD, relative antibody concentration and epitope density, as well. The special value for the characterization of epitope density is the concentration of the bound antibody at the intersection of the other two special values, that is, it is the concentration of bound antibody when the concentration of both the free antigen and the serum antibody is equal to the KD. We call this value standard epitope density, and it is the concentration of epitopes observed under standard conditions.

The range of displayed values should be chosen so as to allow the positioning of the obtained prominent values and also to highlight whether the obtained values are in a common range. We propose the use of a 3D display with the undiluted serum being the end point of the serum antibody concentration range (log[Ab]_s_) and the plane corresponding to this value facing the observer. There are two advantages of this arrangement: (1) the visible right face of the display cube depicts the behavior of undiluted serum at different antigen densities; and (2) if the concentration of antibodies is below the estimated KD, the line corresponding to the KD derived from serum dilution moves “out” of the gradated display range. This implies that the KD is obtained by extrapolation and that the ability of antibodies to saturate the target antigen is weak (Figure 3). Another indication of low relative antibody concentration is the separation of the line indicative of epitope density from the 3D surface. The concentration of Ag-binding Abs in serum relative to KD ([Ab]_s_/KD) is sometimes referred to as effective or thermodynamic concentration and determines the extent of Ag saturation. The position of the red line, given by log([Ab]s/KD), is therefore an indication of the ability of serum Abs to saturate Ag.

### 3.2. Application of the Visualization Scheme to Experimental Data

#### 3.2.1. Demonstration of Individual Differences

To demonstrate the visual impressions of binding differences between two seropositive individuals with anti-SARS-2-CoV antibodies we selected two samples from a previously published dataset with marked differences in KD, concentration and epitope density (Figure 4). Sample PS604 contained antibodies with an antibody concentration close to the KD value, a KD in the middle of the observation range and a relatively high standard epitope density. Sample PS609 exhibited higher affinities (blue line shifted towards lower values) and greater relative concentrations (red line shifted towards higher titers), yet lower epitope densities (purple line shifted downwards).

#### 3.2.2. Demonstration of Group Differences

Besides highlighting differences in the binding properties of individuals, characteristic differences between groups can also be demonstrated using the 3D surface display. The key variables of affinity (KD), relative concentration ([Ab]_s_/KD) and standard epitope density ([Ab]°_b_) can be averaged using an appropriate statistic (mean or median) and the surface and lines can be generated for each antibody isotype (Figure 5).

Both Figure 4 and Figure 5 suggest that the concentration of bound antibodies is not necessarily directly determined by affinity and concentration but that the contribution from an independent variable is required. In spite of the observed lower affinity (blue line shifted right towards higher values) and lower relative concentration (red line shifted right towards lower titers), for all three immunoglobulin classes, the bound antibody concentrations under reference conditions are higher in the seropositive group (Figure 5). According to our model, that independent variable is the epitope density and its effects contribute to the values along the z axis. When comparing the binding of three different immunoglobulin classes, in spite of the similar values of affinity and relative concentration (blue and red lines), the value of the reference bound Ab concentrations is different (purple lines), again pointing to the contribution of epitope density. Therefore, the z axis may represent the different extent of contributions from different antibody isotypes (compare IgM and IgA versus IgG epitope density in the seropositive group) since the epitope density measured here is the effective density to which antibodies are measurably bound, not just the structural epitope density in the sense of surface patches of molecules.

## 4. Discussion

The reactivity of serum antibodies is often characterized by a single number, usually with arbitrary binding units [10] and less frequently with proper concentration units [25]. While a single number and scale might be suitable for setting diagnostic cut-off values and might be related to protectivity [12], in-depth characterization is required for universal comparability and a quantitative systems biological approach. The use of monoclonal antibodies as molecular references for binding activity [11,12] is undoubtedly useful for the standardization of measurements [26] but can only yield concentrations expressed in monoclonal antibody equivalent units, applicable to particular antigens. The generalization of measurements to all antigens and antibody isotypes and the generation of antigen shape spaces and antibody-binding data landscapes for quantitative systems biological databases would require the universal measurement units of biochemistry. We propose that the measurement of the three key variables of serum antibody binding has the potential to reveal as yet unknown correlations with clinical immunological characteristics of health and disease. These three variables, KD, relative concentration [Ab]/KD and standard epitope density [Ab]b° characterize affinity, thermodynamic concentration and Ag valency. While such biochemical variables can also be measured by technologies different from the cited dual-titration microspot assay [13,20,21], we suggest the use of platforms that simultaneously measure them, because separate measurements—especially using separate methods—can introduce methodological bias and variation into each variable, unlike the simultaneous estimation of the three. Nevertheless, if values for all three variables are available, the antibody binding in 3-concentration display, Ab3cD, can be used generally for visualization.

While simply listing three values may reveal to an expert eye the nature of alterations in antibody reactivity, a proper visualization technique can help to identify the extent of the contributions of those three factors while still conveying the complexity of binding. The titration of two parameters can be effectively displayed in a three-dimensional space; what is also important is the arrangement of the axes, the setting of ranges for visualization and the self-explanatory esthetics. We choose an angle of rotation for the 3D surface so that it slopes down towards the left back corner so the viewer can look over the surface and that the behavior of the undiluted serum faces the viewer (Figure 2, Figure 3, Figure 4 and Figure 5). While several serological protocols use a predefined serum dilution for the measurement, the titration of serum allows us to explore the binding of different concentrations of antibodies. In fact, it is the undiluted serum that interacts with blood-borne antigens in the body, so this is highly relevant information. Changing antigen density at fixed undiluted serum antibody concentrations reveals the combined effects of KD and relative concentration on the right front face of the diagram. The whole range of tested serum antibody concentrations is displayed starting from undiluted serum, as discussed above. The antigen concentration range is the tested range itself and should contain the estimated KD, which is therefore obtained by interpolation.

Strictly speaking, all the estimated values for the three variables are apparent or effective values: effective standard epitope density, effective concentration and effective KD, since these are all influenced by interactions ‘invisible’ to the measurement method, like antibody isotypes not detected but still binding and masking epitopes. These values are measurable as the outcome of many interactions in a complex system as opposed to a simple bimolecular interaction under clean experimental conditions.

We introduce here a technical term, epitope density, which is the number of antigen molecule surface patches available for binding by antibody paratopes in a unit area as determined by the number of antigen molecules per unit area and the number of epitopes per antigen molecule. Epitope density values expressed as the concentration of antibody binding sites are obtained from signal intensities via calibration [23]. In turn, standard epitope density, the number of epitopes bound to antibodies under selected equilibrium conditions (standard conditions), will be determined by the affinity and concentration of the antibodies (serum dilution). Unlike epitope density, which depends on the experimental conditions, standard epitope density is a quality of the serum. Counterintuitively a lower standard epitope density may be accompanied by stronger binding forces at a given relative concentration, since higher affinity means lower KD values and standard epitope density values comprise both KD and molecular epitope density:(2)$${\left[Ab\right]}_{b}^{\circ }=K_{D}*{\left[epi\right]}_{m}$$

Conversely, the higher standard epitope density may indicate higher molecular epitope density and therefore greater chemical potential of the antigen with respect to the antibody isotype being measured. This is revealed, for example, by the equilibrium process with three epitopes per antigen molecule, which therefore binds three antibodies according to the equation(3)$$Ag+3Ab⇌AgAb_{3}$$
with potentially lower affinity. We can better understand this phenomenon if we think about decreasing the average affinity of antibodies. This means that we approach the overall average binding affinity of all serum antibodies and we measure the binding of all antibodies. Even though we obtain a high signal, this is generally considered non-specific low-affinity binding with no immunological and clinical relevance. Immunoassays specifically tailored for diagnostic measurements avoid this by using low antigen concentrations, practically excluding low-affinity interactions from the measurement [17]. Single-point immunoassays, however, may also mistakenly qualify high-affinity binding samples as negative, because the concentration of bound antibodies is low. Systemic autoimmune diseases with cycling disease activity (periods of relapse and remission [27,28,29,30]) may also show periods of seropositive and seronegative status, in spite of the gradual affinity maturation of autoantibodies.

## 5. Conclusions

In summary, the proposed visualization scheme answers (1) how strongly antibodies are binding, (2) whether antibodies are present in excess (3) and how many sites antibodies are directed against. The answers are the effective KD, the relative antibody concentration and the standard epitope density, each represented by a line (Figure 2, Figure 3, Figure 4 and Figure 5) and all the three contributing to the routinely observable fractional occupancy that is deconvoluted by the dual-titration assay. Such in-depth analysis may not be required for routine diagnostic tests but should be useful for quantitative and systems biological studies where serum antibody responses are involved.

## Figures and Tables

**Figure 1 antibodies-14-00068-f001:**
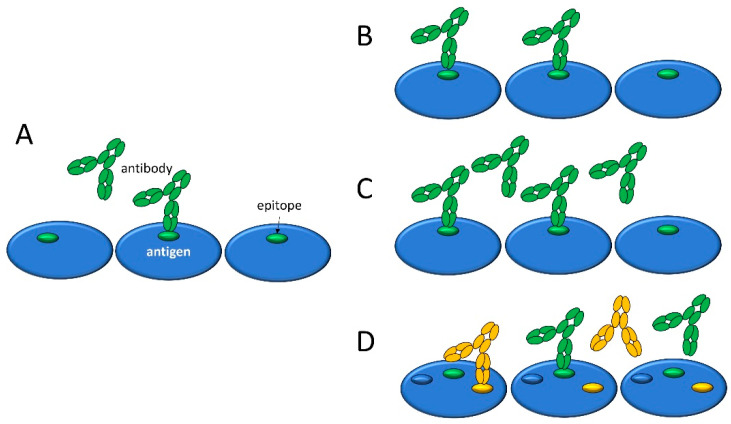
Schematic representation of the key immune-response-related factors that determine the extent of antibody binding to surface-immobilized antigens. The number of antigen-bound antibody molecules per unit area in equilibrium (**A**) increases when antibody affinity (**B**) and/or concentration (**C**) and/or the number of recognized epitopes per antigen molecule, antigen epitope density (**D**) increases. Green and orange colors stand for distinct antibody clones with different cognate epitopes.

**Figure 2 antibodies-14-00068-f002:**
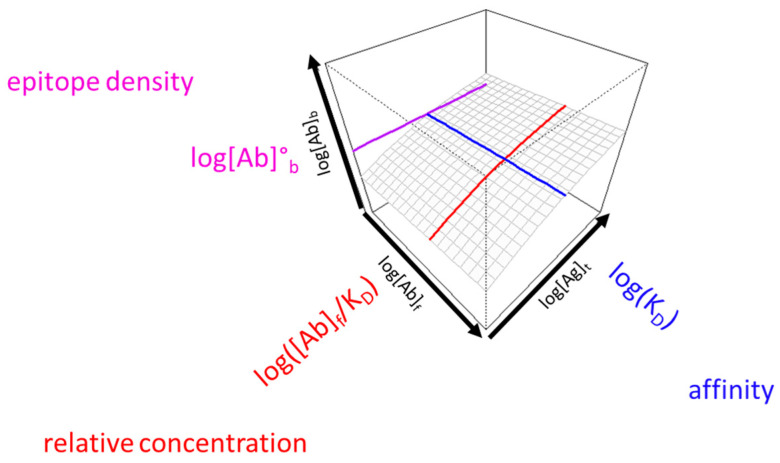
Interpretation of the 3D binding surface. The surface corresponds to the logarithm of the concentrations of bound antibodies (log[Ab]_b_) obtained by titrating binding sites immobilized on the surface (log[Ag]_t_) and antibody concentrations (log[Ab]_f_). The contributions to binding of the three factors, namely affinity, concentration and epitope density, in a given serum sample are characterized by the three lines (blue, red and purple, respectively), identifying special values along the three axes. Subscript letters stand for bound, b; total, t; free, f.

**Figure 3 antibodies-14-00068-f003:**
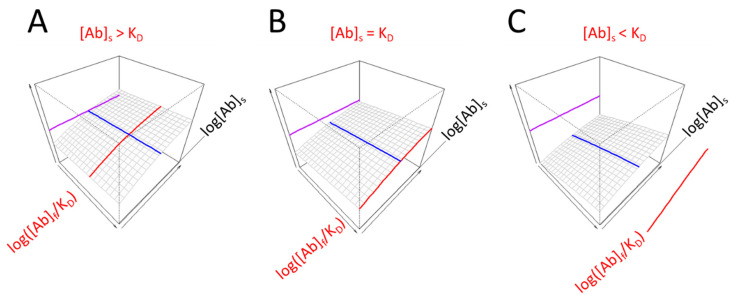
Visual effects of antibody concentration change relative to KD. The bottom right face depicts undiluted serum binding. The KD estimated from serum dilution (red line) is smaller (**A**), equal to (**B**) or greater than (**C**) the apparent serum antibody concentration. Epitope density (purple) and KD (blue) values are identical for the three diagrams. Subscript s stands for serum and log[Ab]_s_ is the logarithm of serum Ab concentration in undiluted serum.

**Figure 4 antibodies-14-00068-f004:**
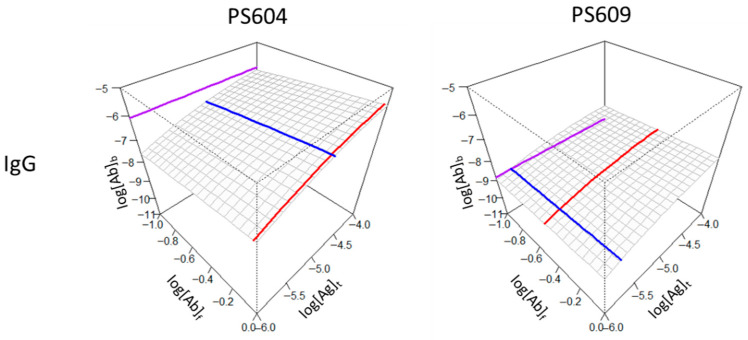
Visualization of individual differences in the seropositive group for IgG reactivity. A serum sample with a low relative concentration and lower affinity (PS604) can still show a higher epitope density than a sample with a higher relative concentration and higher affinity (PS609). Axes and color codes are identical to previous figures.

**Figure 5 antibodies-14-00068-f005:**
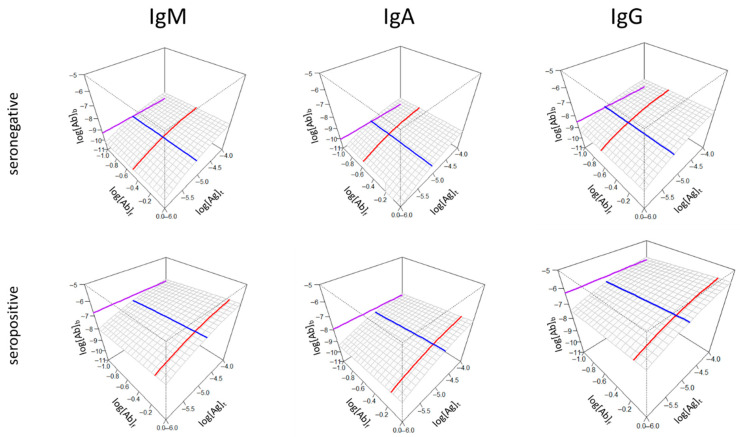
Effects of relative antibody concentration changes for three antibody classes (IgM, IgA, IgG) in two patient groups (seronegative and seropositive). In spite of the higher standard bound antibody concentrations in the seropositive group, indicated by the purple line positions, antibody concentrations relative to KD are higher in the seronegative group (red line), suggesting the targeting of more epitopes by antibodies during an active immune response. Axes and color codes are identical to previous figures.

## Data Availability

Visualized data are contained within the Appendix A.

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
