# Peer review of "A 3D Surface Plot for the Effective Visualization of Specific Serum Antibody Binding Properties"

_2073-4468, 2025, doi:10.3390/antib14030068_

Round 1

Reviewer 1 Report

Comments and Suggestions for Authors

Review for the journal Antibodies of manuscript "A 3D surface plot for the effective visualization of specific serum antibody binding properties" by Prechl et al.

  This paper presents a novel graphical approach for visualizing serum reactivity to specific antigens. The method is valid and likely to be useful, so it merits publication. However, the value of the paper could be greatly improved by correcting the few flaws listed below.

   The Abstract faults other assays for generating only a single variable, without saying what it is. Then it promises to deliver "the essential biochemical variables" and "key variables", without saying what they are, or even how many variables it considers essential/key. Please list them here. Is one of the promised set of variables the same as, or corresponding to, the "single variable" that other assays provide? If so, this information will help to explain how the new method builds on and surpasses the other assays.  

   Line 44. "average affinity" is not quite correct. Suppose a sample with normal antibodies was spiked with an equal number of nonbinding, zero-affinity antibodies. This would reduce to half the average affinity of the antibodies present, but it would not decrease the readout to half. The measured readout is more like "aggregate affinity". A central point to be considered explicitly is that measured titers typically result from a few strong binders plus a large number of weak binders.

   Line 84. "Tables containing the visualized dataset are available in Supplementary data." The materials received by this reviewer include only one table, and it contains only 18 data values. Which Figures does this dataset correspond to? And which is meant by 'the visualized dataset'? All the graphs? Since all the graphs show surfaces that appear to be based on grids of multiple measurements, it is hard to understand even one of them could be based on such a short spreadsheet.

   Lines 91-96. Considering that it is central to the paper, this sentence seems far too long and dense, with several mathematical unclarities:  (1) Every point on the surface is defined by three coordinates, so it is confusing to read that z defines the surface coordinates, i.e., defines them all. Does z correspond to the vertical axis of the plot? And do x and y likewise correspond to plot axes, per Cartesian convention? If so, readers will benefit from being given this info explicitly, perhaps marked on the graph, also indicating the 'origin' with a dot or circle. (2) "the point of intersection of [two] inflection points" doesn't make sense. A point may be defined as the intersection of two lines, but not as the intersection of two points. (3) "two Richards functions". Which two? Since the Richards function has barely been mentioned til now, and is central to the paper, this is too cursory. Please give readers (perhaps within a previous paragraph) a 1 or 2 sentence summary of the relevant role of the Richards function, including the key significance of its inflection point. (4) In contrast to all that, it is needless to say what log stand for.

   Line 184. "measurement of three key variables". Either here or earlier in the paper, please give a clear inventory of these three measurements. Are they distinct types of measurements, or are they all fluorescence intensity, under three different experimental perturbations?

   Line 211. Please give the practical units of epitope density. Readers will more readily accept this introduced term if given some values or a typical range for this key number. With earlier graphs, where epitope density is derived, give the derived value. Also, in a sentence or two, how does its surface concentration relate to the operational variable of the solution concentration of antigen?

Author Response

Q1, The Abstract faults other assays for generating only a single variable, without saying what it is. Then it promises to deliver "the essential biochemical variables" and "key variables", without saying what they are, or even how many variables it considers essential/key. Please list them here. Is one of the promised set of variables the same as, or corresponding to, the "single variable" that other assays provide? If so, this information will help to explain how the new method builds on and surpasses the other assays.  

A1, >> We have complemented the abstract with the information proposed by the reviewer, listing the variables that are essential for biochemical characterization of antibody binding and their relationship with conventional assay results.

Q 2,  Line 44. "average affinity" is not quite correct. Suppose a sample with normal antibodies was spiked with an equal number of nonbinding, zero-affinity antibodies. This would reduce to half the average affinity of the antibodies present, but it would not decrease the readout to half. The measured readout is more like "aggregate affinity". A central point to be considered explicitly is that measured titers typically result from a few strong binders plus a large number of weak binders.

A2, >> We agree with the reviewer that average, interpreted as arithmetic mean, would not properly convey what we intend to say about the assay. There are some theoretical points here that we would not like to detail in the manuscript. We assume that the distribution of antibody concentration as a function of affinity, sometimes referred to as affinity distribution, is not Gaussian but exponential. The huge number of very low affinity binders extends into a tail of high affinity binders. The expected value of this distribution is determined by the rate of the distribution in the exponent and indeed reduces the value of average or aggregate affinity. In that sense we do consider the contribution of all binders just as proposed by the reviewer. In a similar manner, the interpretation of concentration is also interpreted according to this model and is not a concentration associated with the binder with most frequent affinity but rather an expected value from the distribution. We have added the suggested expression to the test to indicate this subtlety.

Q3,   Line 84. "Tables containing the visualized dataset are available in Supplementary data." The materials received by this reviewer include only one table, and it contains only 18 data values. Which Figures does this dataset correspond to? And which is meant by 'the visualized dataset'? All the graphs? Since all the graphs show surfaces that appear to be based on grids of multiple measurements, it is hard to understand even one of them could be based on such a short spreadsheet.

A3, >> We thank the reviewer for pointing out that the shared dataset cannot be associated with the figures. We have identified the data by naming the excel sheets according to figures. The values shown in the sheets were visualized by the program in the supplementary file using the mathematical expression described under Materials&Methods. Thus, each diagram requires the values of three variables only to generate the 3D surface using the double Richards function.

Q4,   Lines 91-96. Considering that it is central to the paper, this sentence seems far too long and dense, with several mathematical unclarities:  (1) Every point on the surface is defined by three coordinates, so it is confusing to read that z defines the surface coordinates, i.e., defines them all. Does z correspond to the vertical axis of the plot? And do x and y likewise correspond to plot axes, per Cartesian convention? If so, readers will benefit from being given this info explicitly, perhaps marked on the graph, also indicating the 'origin' with a dot or circle. (2) "the point of intersection of [two] inflection points" doesn't make sense. A point may be defined as the intersection of two lines, but not as the intersection of two points. (3) "two Richards functions". Which two? Since the Richards function has barely been mentioned til now, and is central to the paper, this is too cursory. Please give readers (perhaps within a previous paragraph) a 1 or 2 sentence summary of the relevant role of the Richards function, including the key significance of its inflection point. (4) In contrast to all that, it is needless to say what log stand for.

A4, >> We have completely rephrased and extended this section to properly introduce mathematical backgrounds of the visualization to the reader.

Q5,   Line 184. "measurement of three key variables". Either here or earlier in the paper, please give a clear inventory of these three measurements. Are they distinct types of measurements, or are they all fluorescence intensity, under three different experimental perturbations?

A5, >> The visualization tool we present here builds on the measurement of affinity, relative concentration and epitope density data. Estimates for these three variables are obtained from a single multiplex measurement that generates a data matrix, from which the values are obtained by fitting the Richards curves to the measurement data. This is now explained in more detail under the methods and discussion sections as well.

Q6,   Line 211. Please give the practical units of epitope density. Readers will more readily accept this introduced term if given some values or a typical range for this key number. With earlier graphs, where epitope density is derived, give the derived value. Also, in a sentence or two, how does its surface concentration relate to the operational variable of the solution concentration of antigen?

A6, >> We introduce the term epitope density to account for variability in bound antibody concentration that is not explained by affinity and concentration. Technically, as stated by the reviewer, it is a surface concentration calculated from the relationship between the signal of immobilized immunoglobulins of the respective isotype and its known concentration in the solution from which it is immobilized. Such a calibration can provide only a surrogate solution concentration value, expressed in common molar concentrations (like on the vertical axis in the figures) but is still reliable for comparative measurements of serum antibody binding. Using those values as surrogate solution concentrations, epitope density will be the concentration of antibody binding sites and as multiple of antigen concentration and molecular epitope density. Since bound antibody concentration can only be fraction of this value, determined by the fractional saturation value, epitope density is always higher than bound antibody concentration. Additionally, epitope density effectively available for a detected antibody class is lower than total epitope density because of common targets and competition between antibody classes. Under standard conditions, which is defined in our model as in the presence of free antigen and free antibody concentrations equal to the KD, the measured and calculated concentration of bound antibodies provides a measure of effectively available binding sites – influenced by all the listed factors – and is shown in the figures as the value [Ab]b°.

Reviewer 2 Report

Comments and Suggestions for Authors

The work of Prechl et al. is devoted to the development of a 3D method for visualization of variables of the main parameters of serum antibody-antigen interaction (Kd, relative antibody concentration and epitope density). As a key unique feature of the proposed method, the ability to account for epitope density changes is stated. The authors used a commercial dataset obtained by screening sera from COVID-19 convalescents to demonstrate the applicability of the proposed modeling method to illustrate the properties of individual samples and experimental groups. Although the work has practical relevance, it has a number of shortcomings, which are listed below and need to be addressed.

First of all, since the article is quite short and poor in scientific data, I think the best form for it would be Brief Report.

In the abstract (lines 18-19) it is stated that the proposed method, unlike conventional assays, allows showing a change in epitope density, while it is not clear where this is demonstrated in the paper and there is no such subchapter in the results.

The interrogative sentences (lines 37-38, 242-244) should be rephrased.

Lines 45-47: should be expanded as not all standard approaches are exemplified, e.g. ELISA and SPR methods are not mentioned.

Lines 50-52 and 178-181: the proposed imaging method does not solve the stated problem, so it is not clear why the authors speculate about it.

In the introduction, the concept of equilibrium should be clarified and lines 205-219 should be moved, as it is not clear why exactly the parameters are considered.

It should be clarified exactly how many serum samples were studied and what technique was used to obtain data from the screening of sera of the convalescents (lines 82-86): method, diagnostic kit, equipment, etc. Although the calculation of logC, 𝜈 and logxi a was referenced as [14, 15], it is necessary to briefly describe it here. And by the way, how were obtained the data for ‘group’ samples?

Lines 93-94: the meaning and method of defining the parameter 𝜈 are unclear.

Line 102: actually, the meaning of the three parameters was not disclosed above, and this point should definitely be added to the Introduction.

Figure 2: the caption does not explain what log[Ag]f is and what the degree sign at log[Ab]b means.

Line 130 ff: the authors suggest using undiluted serum parameters, but standard test systems use a series of dilutions. The applicability of the visualization method to real tests should be explained, and vice versa, approach should be adapted for diluted sera.

Line 145: is this a subheading?

Line 160: the [Ab]s/KD parameter has not been mentioned before.

How do the authors account for statistical tests in demonstrating intergroup differences? If Figure 5 visualizes the parameters of individual sera, these are merely representative examples, but not a comparison of experimental groups.

Lines 242-244: the proposed visualization method operates with previously obtained experimental data that already answer these questions; it only presents them visually. The value of the study should not be exaggerated.

Author Response

Q1, First of all, since the article is quite short and poor in scientific data, I think the best form for it would be Brief Report.

A1, >> We thank the reviewer for highlighting this option for the publication.

Q2, In the abstract (lines 18-19) it is stated that the proposed method, unlike conventional assays, allows showing a change in epitope density, while it is not clear where this is demonstrated in the paper and there is no such subchapter in the results. A2, >> We have revised figure 1 and the text regarding the explanation of our approach towards measuring and interpreting epitope density. In the revised text lines 185-192, 212-220 are related to this concept.

Q3, The interrogative sentences (lines 37-38, 242-244) should be rephrased.

A3, >> We have rephrased the sentences.

Q4, Lines 45-47: should be expanded as not all standard approaches are exemplified, e.g. ELISA and SPR methods are not mentioned.

A4, >> We have rephrased this sentence to emphasize what we intended to say here: common technologies can generally assess biochemical properties of serum antibodies only one-by-one, and usually following elaborate preanalytical procedures. Recently developed technologies are focusing on the simultaneous determination of these biochemical variables in a single assay. The advantage of such approaches is also mentioned in lines 214-221.

Q5, Lines 50-52 and 178-181: the proposed imaging method does not solve the stated problem, so it is not clear why the authors speculate about it.

A5, >> We agree that the solution to standardized quantitative antibody serology is not the visualization method presented here. We refer to others’ and to our own method as potential solutions to understanding the complex nature of polyclonality, polyspecificity of serum antibodies that can provide quantitative biochemical characterization, and the presented visualization tool we think is useful for conveying the complexity while also identifying values of variables responsible for serum antibody reactivity.

Q6, In the introduction, the concept of equilibrium should be clarified and lines 205-219 should be moved, as it is not clear why exactly the parameters are considered.

A6, >> Let us kindly disagree with the reviewer here, as the considered parameters are the essence of the article and we would not like to remove the paragraphs that introduce these parameters. Why these parameters are considered essential and how they are calculated and interpreted are now better explained in the revised form of the manuscript.

Q7, It should be clarified exactly how many serum samples were studied and what technique was used to obtain data from the screening of sera of the convalescents (lines 82-86): method, diagnostic kit, equipment, etc. Although the calculation of logC, ? and logxi a was referenced as [14, 15], it is necessary to briefly describe it here. And by the way, how were obtained the data for ‘group’ samples?

A7, >> We have given a more detailed description of the mathematical part of the visualization approach in the Methods section.

Q8, Lines 93-94: the meaning and method of defining the parameter ? are unclear.

A8, >> The meaning and method of obtaining parameter ? are now explained in the Methods section, along with the other variables and the parametrization of the Richards function.

Q9, Line 102: actually, the meaning of the three parameters was not disclosed above, and this point should definitely be added to the Introduction.

A9, >> In the revised manuscript we name the biochemical parameters and their meaning already in the abstract.

Q10, Figure 2: the caption does not explain what log[Ag]f is and what the degree sign at log[Ab]b means.

A10, >> The legend was corrected to explain the meaning of subscripts.

Q11, Line 130 ff: the authors suggest using undiluted serum parameters, but standard test systems use a series of dilutions. The applicability of the visualization method to real tests should be explained, and vice versa, approach should be adapted for diluted sera.

A11, >> It is only the visualization that depicts the estimated behavior of undiluted serum. We agree with the reviewer that the use of undiluted serum generally results in background and noise to an extent that results are not reliable. The method that generates the data for the proposed visualization plot actually uses series of dilutions and fits generalized logistic curves to the experimental data. The reason we think the display of estimated ideal behavior of undiluted serum is useful because it is the biologically relevant in vivo behavior of serum antibodies, which is more appropriate to compare than activity values obtained from different serum dilutions that were selected simply for optimizing diagnostic assays.

Q12, Line 145: is this a subheading?

A12, >> We thank the reviewer for pointing out that this subheading was not properly formatted, we corrected and renumbered the subheadings.

Q13, Line 160: the [Ab]s/KD parameter has not been mentioned before.

A13, >> We agree that even though relative concentration is introduced earlier, the expression [Ab]s/KD should also be explained in previous sections. We have added explanations for [Ab]s, [Ab]s/KD and log([Ab]s/KD) to relevant parts of the text and legends.

Q14, How do the authors account for statistical tests in demonstrating intergroup differences? If Figure 5 visualizes the parameters of individual sera, these are merely representative examples, but not a comparison of experimental groups.

A14, >> We agree that the proposed visualization method does not address statistical differences. It only serves to demonstrate in what parameters, and therefore in what biochemical properties the studied serum samples appear different. The statistical characterization of that difference and the indication of statistical significance can follow conventions and was not the aim of this article.

Q15, Lines 242-244: the proposed visualization method operates with previously obtained experimental data that already answer these questions; it only presents them visually. The value of the study should not be exaggerated.

A15, >> We would not like to exaggerate the value of the visualization method. What we intended to tell the reader is that in-depth analysis (by whatever means) is required for systems biological studies and the proposed visualization scheme is suitable for displaying results from an in-depth biochemical analysis of serum antibody binding. The main intended message of our article is that serum antibody responses are better represented by three parameters instead of the usual single parameter, and these three parameters are ideally visualized in three dimensions along with the fitted curves appearing as a 3D surface. Because of the complexity of the mathematical function used for fitting we assumed that a paper dedicated to the visualization would help the scientific community better understand purely experimental papers.

Reviewer 3 Report

Comments and Suggestions for Authors

In the manuscript entitled ¨A 3D surface plot for the effective visualization of specific serum antibody binding properties¨, " the authors claim a new scheme of visualization for representing individual and group epitope density changes and polyclonal antibodies interaction. This scheme considered the affinity, relative concentration, and epitope density.

Nevertheless, some issues need clarification before acceptance. These critical issues must be addressed to improve the quality of the manuscript for publication. 

  1. A clearer representation is necessary in Figure 1 for clarity; authors can use scientific software to create a new figure that better illustrates the antigen-antibody equilibrium.
  2. Authors indicate the use of human samples for PS604 and PS609; however, there is no declaration of acceptance of the clinical protocol, nor is a corresponding number provided. Those samples were obtained from SARS patients, making it mandatory to declare the clinical protocol acceptance number.
  3. The authors only indicate the IgA, IgM, and IgG for two samples (Fig. 5). Considering the claim that the emergence of antibody clones brings about ¨changes in epitope density ¨, how do the authors demonstrate the epitope density for these samples? How is the antibody diversity in these samples? Do the authors perform some experimental assays?
  4. In other published manuscripts by the same authors (doi: 10.3389/fimmu.2025.1494624), they report a 2D scheme very similar to the 3D scheme proposed in this manuscript, also with anti-SARS antibodies. For this reason, the scientific soundness of the manuscript submitted to this Journal is low.
  5. In lines 166-167 author claims  ¨suggests that the concentration of bound antibodies is not necessarily directly determined by affinity and concentration but is an independent variable¨. Do the authors have any experimental data that supports this formation? Please include these assays in the supplementary material.
  6. The conclusion section is missing. The authors only include a summary and limitations of the studies. They need to include a conclusion based on the results obtained.
  7. Further analysis of additional serum samples is necessary. 
  8. Please explain how the new scheme proposed in this manuscript could impact (support) the statistical significance of results obtained by other research groups that want to report their latest findings using this new scheme. 

In summary, although the authors defend the work as a relevant step in the antigen-antibody equilibrium, they present few results to demonstrate the scientific contribution of this work and to demonstrate the relevance and activity of the proposed 3D scheme. Further analyses, such as those suggested here, are necessary to enhance the understanding of the research's importance and to elucidate the advantages that authors claim.

Author Response

Q1, A clearer representation is necessary in Figure 1 for clarity; authors can use scientific software to create a new figure that better illustrates the antigen-antibody equilibrium.

A1, >> We improved the quality of the figure by replacing symbols for antibodies, by improving color scheme and by complementing the legends.

Q2, Authors indicate the use of human samples for PS604 and PS609; however, there is no declaration of acceptance of the clinical protocol, nor is a corresponding number provided. Those samples were obtained from SARS patients, making it mandatory to declare the clinical protocol acceptance number.

A2, >> The serum samples were obtained commercially and not collected for the purpose of the study. Commercially available clinical samples are generally exempt from declaring protocols and declarations. We indicated this under section 2.1.

Q3, The authors only indicate the IgA, IgM, and IgG for two samples (Fig. 5). Considering the claim that the emergence of antibody clones brings about ¨changes in epitope density ¨, how do the authors demonstrate the epitope density for these samples? How is the antibody diversity in these samples? Do the authors perform some experimental assays?

A3, >> We did not perform experimental assays for this article; we used data from previously performed experiments. The observation of changed epitope density is a deduction from the analysis of results and is related to the relative values of the three descriptors of binding. Comparable affinity and concentration values in two samples accompanied by large differences in the concentration of bound antibodies can only be attributed to differences in the density of epitope detected in the measurement. In fact, both figure 4 and 5 shows examples of lower affinity and lower relative concentration being accompanied by increased bound antibody signals, which can only be explained by differences in epitope density in our model.

Q4, In other published manuscripts by the same authors (doi: 10.3389/fimmu.2025.1494624), they report a 2D scheme very similar to the 3D scheme proposed in this manuscript, also with anti-SARS antibodies. For this reason, the scientific soundness of the manuscript submitted to this Journal is low.

A4, >> We thank the reviewer for drawing our attention to any similarities with previous published material. The 2D schemes applied previously indeed visualize fluorescent signals as a function of antigen density and antibody dilution. However, the exact purpose of this article is to highlight and explain why a 3D visualization is required for the efficient visualization of results that comprise 3 parameter values. We realized that by showing those values in two distinct diagrams we miss the important relationship of the combined effects of antigen and antibody densities on the signals, and that by combining those two diagrams we can represent combined effects on the bound antibody concentrations. In the revised manuscript several changes in the text focus on explaining this phenomenon.

Q5, In lines 166-167 author claims ¨suggests that the concentration of bound antibodies is not necessarily directly determined by affinity and concentration but is an independent variable¨. Do the authors have any experimental data that supports this formation? Please include these assays in the supplementary material.

A5, >> The reviewer is correct to point out that this sentence is misleading. The concentration of bound antibodies is not an independent variable but rather factors contributing to the concentration of bound antibodies should include an independent variable, otherwise comparable affinity and concentration should result in comparable signals of bound antibodies. We corrected the text accordingly.

Q6, The conclusion section is missing. The authors only include a summary and limitations of the studies. They need to include a conclusion based on the results obtained.

A6, >> We separated the last paragraph of discussion to create a conclusion section.

Q7, Further analysis of additional serum samples is necessary.

A7, >> We agree that further analysis of additional samples will provide further proof for the utility and applicability of the visualization method. In this article we only intend to introduce the theory behind this approach, provide proof-of-principle and examples for the visualization method. We propose the publication of this manuscript as a Brief Report.

Q8, Please explain how the new scheme proposed in this manuscript could impact (support) the statistical significance of results obtained by other research groups that want to report their latest findings using this new scheme.

A8, >> To report results using this new scheme researchers have to carry out measurements that provide the three parameters identified by our model as key for full serum antibody binding characterization. While these three factors can also be measured by technologies different from the cited dual-titration microspot assay, we suggest the use of this platform because independent measurements can introduce methodological bias and variation into each variable, unlike the simultaneous estimation of the three. Our visualization scheme does not address statistical significance, we suggest to carry out conventional statistical assays and report significance following conventional means.

Round 2

Reviewer 2 Report

Comments and Suggestions for Authors

It seems that some of the required details have been introduced to the text. However, I still recommend reclassifying the article in the Brief Report form as poor in novel scientific data and extending the Introduction with the description of three parameters and equilibrium state as it was suggested previously.

Author Response

Q1: It seems that some of the required details have been introduced to the text. However, I still recommend reclassifying the article in the Brief Report form as poor in novel scientific data and extending the Introduction with the description of three parameters and equilibrium state as it was suggested previously.

A1: We expanded the Introduction section with further description of the terms affinity, concentration and equilibrium.

Reviewer 3 Report

Comments and Suggestions for Authors

In this new manuscript version, the authors respond to all reviewer suggestions. Overall, the manuscript was improved. The suggestion of making the manuscript a Brief report is appropriate and under the content.

The improvement of Figure 1 and the inclusion of a better description of Figures 4 and 5 make clear the findings and applications of the proposed visualization scheme by the authors.

Author Response

Q1: The improvement of Figure 1 and the inclusion of a better description of Figures 4 and 5 make clear the findings and applications of the proposed visualization scheme by the authors.

A1: We supplemented the descriptions of Figs 4 and 5 so that their relationship to Fig 1 is clearer.